# Nutrition, Overweight, and Cognition in Euthymic Bipolar Individuals Compared to Healthy Controls

**DOI:** 10.3390/nu14061176

**Published:** 2022-03-11

**Authors:** Bernd Reininghaus, Nina Dalkner, Christiane Schörkhuber, Eva Fleischmann, Frederike T. Fellendorf, Michaela Ratzenhofer, Alexander Maget, Martina Platzer, Susanne A. Bengesser, Adelina Tmava-Berisha, Melanie Lenger, Robert Queissner, Elena M. D. Schönthaler, Eva Z. Reininghaus

**Affiliations:** Department of Psychiatry and Psychotherapeutic Medicine, Medical University Graz, 8036 Graz, Austria; berndreininghaus@web.de (B.R.); chr.schoerkhuber@gmail.com (C.S.); eva.fleischmann@medunigraz.at (E.F.); frederike.fellendorf@medunigraz.at (F.T.F.); Michaela.ratzenhofer@t-online.de (M.R.); alexander.maget@medunigraz.at (A.M.); martina.platzer@medunigraz.at (M.P.); susanne.bengesser@medunigraz.at (S.A.B.); adelina.tmava@medunigraz.at (A.T.-B.); melanie.lenger@medunigraz.at (M.L.); robert.queissner@medunigraz.at (R.Q.); elena.schoenthaler@medunigraz.at (E.M.D.S.); eva.reininghaus@medunigraz.at (E.Z.R.)

**Keywords:** bipolar disorder, overweight, cognitive function, nutrition

## Abstract

Bipolar disorder (BD) is associated with impairments in cognitive functions, in which metabolic factors, e.g., overweight, seem to play a significant role. The aim of this study was to investigate the association between nutritional factors and cognitive performance in euthymic individuals with BD. A study cohort of 56 euthymic individuals with BD was compared to a sample of 53 mentally healthy controls. To assess cognitive function, the following tests were applied: California Verbal Learning Test, Trail Making Test A/B, d2 Test of Attention-Revised, and Stroop’s Color-Word Interference Test. Furthermore, a 4-day food record was processed to evaluate dietary intake of macronutrients, specific micronutrients, and food diversity. Body mass index and waist to height ratio were calculated to assess overweight and central obesity. Results showed no nutritional differences between individuals with BD and controls. Individuals with BD performed worse in the d2 test than controls. Hierarchical regression analyses yielded no association between cognitive and nutritional parameters. However, waist to height ratio was negatively correlated with almost all cognitive tests. Central obesity seems to affect cognitive functioning in BD, while the lack of finding differences in nutritional data might be due to problems when collecting data and the small sample size. Consequently, further studies focusing on objectively measuring food intake with adequate sample size are needed.

## 1. Introduction

Bipolar disorder (BD) is associated with impairment in cognitive function not only during acute mood episodes but also during periods of euthymia. Emerging evidence indicates that in comparison with healthy controls (HC), euthymic bipolar patients show impairments in various cognitive domains [1,2,3,4,5,6,7,8,9,10]. These impairments in attention, memory, and executive function are highly relevant as they are determinants of psychosocial and occupational functioning outcomes [11].

A further problem, leading to increased mortality, reduced quality of life, and lower psychosocial and cognitive functioning in BD, is the high rate of somatic comorbidities. In this context, overweight and obesity are well known to be frequently prevalent in patients with mood disorders and especially in BD [12,13,14]. Patients with BD are more often obese than HC, and antipsychotic drugs, which are usually prescribed for the treatment of BD, seem to be associated with an increased likelihood of obesity [14]. Nevertheless, an increased prevalence of overweight and obesity has also been reported for drug-naïve BD patients [15]. It is of high relevance that several studies found a negative effect of overweight and obesity on cognitive function in euthymic patients with BD [16,17,18]. This implies that the prevalence of overweight and obesity is a factor that further decreases the, in many cases, already reduced cognitive functioning in individuals with BD. In general, a variety of lifestyle factors including reduced physical activity [19] and nutritional factors [20] as well as prevalence of psychiatric symptoms, intake of psychotropic drugs, common neurobiological pathways [21], and stigmatization [22] lead to the increased prevalence of somatic comorbidities in individuals with BD.

The influence of nutritional habits on the development and maintenance of overweight and obesity is well described in previous literature. Even more, the topic of nutrition in mental health has gained increasing interest throughout the last years and is especially relevant regarding the effort to improve overweight/obesity and associated cognitive deficits in individuals with BD. In a review [23], Dauncey suggests a multifactorial pathway from nutrition to brain function. In this model, nutrition is affected, among others, by physical activity, environment, microbiota, social interaction, genetics, and age. These factors are suggested to affect neurotrophic and neuroendocrine factors, which, in turn, influence cell signaling and neural pathways and thereby modulate neuronal functions, like synaptic plasticity and adult neurogenesis, and finally can facilitate brain function like cognition and mental health. In addition, nutritional habits including a combination of polyunsaturated fatty acids, minerals, and vitamins may facilitate medication efficacy and reduction of symptoms of patients with mood disorders [24].

There is evidence that patients with BD tend to eat unhealthily when compared to nutritional guidelines [24], especially they tend to have a higher intake of sugar, fat, and processed meats [25,26]. Furthermore, nutritional habits might directly influence cognitive function independent of psychiatric symptoms. In particular, Mediterranean diet/variety [23,27,28,29], intake of omega-3 fatty acids [20,23,30], and B vitamins [31,32] seem to be beneficial for cognitive function. Since cognitive impairment is a severe and frequent consequence of BD, we wanted to further elucidate the complex relationship between nutritional behavior, overweight, and cognitive dysfunction in euthymic individuals with BD.

We hypothesized, (1) that individuals with BD consume more fat, protein, and carbohydrates, and less beneficial micro- and macronutrients than HC, and (2) that there is a positive relationship between the average intake of micro-and macronutrients, nutritional diversity, and different cognitive variables (attention, memory, executive function).

## 2. Methods

### 2.1. Participants

The study sample consisted of 56 bipolar outpatients from the Department of Psychiatry and Psychotherapeutic Medicine at the Medical University of Graz, Austria, diagnosed with DSM-IV [33]. A HC group without any psychiatric disorders was included, consisting of 53 individuals without a personal or family history (first-degree) of psychiatric diseases. The study was part of the ongoing BIPFAT study at the Department of Psychiatry at the Medical University of Graz, which aims to explore the relationship between BD and obesity, metabolism, lifestyle, and cognitive function. For in-depth information about the study design and preliminary results, see previously published reports [17,34,35,36,37].

Participants had to be of legal age, in a state of euthymia or mild depression at the time of study and had given written informed consent prior to their participation. Euthymia was defined as a Hamilton Depression Rating Scale score < 15 and a Young Mania Rating Scale score < 6. Patients were excluded when presenting with neurological (for example Parkinson’s, or Alzheimer’s disease) or medical (for example inflammatory bowel disease) comorbidities. Furthermore, individuals who were not euthymic (HAMD score ≥ 15, YMRS score ≥ 6) were excluded as well. The study was approved by the local ethics committee (Medical University of Graz, Graz, Austria) in compliance with the current revision of the Declaration of Helsinki, ICH guideline for Good Clinical Practice and current regulations (EK-number: 24–123 ex 11/12).

### 2.2. Anthropometric Measures

Body Mass Index (BMI) was categorized into normal weight (BMI = 18.0–24.9 kg/m^2^), overweight (BMI = 25.0–29.9 kg/m^2^), and obese (BMI ≥ 30 kg/m^2^). Waist to height ratio (WHtR) was calculated based on measured individual waist circumference and height (WHtR = waist/height). More specifically, waist circumference was measured with a tape measure in centimeters at the smallest circumference between the rib cage and the iliac crest. WHtR is more sensitive to detect health risks, e.g., abdominal obesity, than BMI [38]. A cut-off value of 0.5 was used, with a value higher than 0.5 indicating an increased health risk, for example coronary heart disease and metabolic syndrome [38]. In addition, this cut-off value is indicative for men and women as well as for individuals with different ethnic backgrounds.

### 2.3. Psychological Inventories

The Hamilton Depression Scale [39] and Young Mania Rating Scale [40] were conducted as a structured interview by a psychiatrist or a psychologist. According to German guidelines (S3-Leitlinie/NVL, 2012), a HAMD score of 17 or higher indicates a moderate depressive syndrome. The German version of the YMRS has a Cronbach’s α of 0.74 [40].

Furthermore, the self-reported Beck-Depression-Inventory (BDI-II) was used to assess depressive symptoms during the last 2 weeks [41]. According to Hautzinger and colleagues [41], a score of 18 or higher on a scale from 0 to 63 indicates a clinically relevant depression. With a Cronbach’s α of ≥0.84, the scale shows a good internal consistency [42].

To measure verbal learning and memory, the German version of the California Verbal Learning Test (CVLT) [43] was administered. The CVLT enables an individual assessment of verbal learning strategies and memory processes and consists of two wordlists, each with 16 items, respectively, four items from four semantic categories. First, wordlist A is presented five times in total and is immediately followed by a free recall trial after each presentation. After the five learning trials, the interference task (list B) must be recalled once. List B is followed by an immediate free recall and a category-cued recall of List A. Following a 20-min delay during which other non-interfering tests are administered, the free-recall, cued recall, and yes/no recognition memory of List A are tested. In this study, the learning sum (trial 1–5), the short delay free recall, and the long delay free recall were administered. Reliability scores of 0.75 and 0.79 (version 1 and 2) for split-half reliability and 0.60 for retest reliability are satisfactory.

The Trail Making Test consists of two parts, part A (TMT A) and part B (TMT B), which are elaborated successively [44]. To process TMT A, participants have to connect circled numbers starting from the number one in ascending order to the number 25 as quickly and accurately as possible. Processing time is measured in seconds and represents the individual attention and psychomotor speed [7,45]. The TMT A is widely used to measure psychomotor processing speed. Part B, TMT B, is processed similarly to TMT A, except that participants must connect circled numbers and letters alternately in ascending order. Processing time evaluates individual cognitive flexibility, as well as visual motor and visual spatial abilities, and executive functions [7,45]. However, to control strategic speed factors and visual complexity factors, difference derived scores (TMT B minus TMT A) were calculated [46] as a more sensitive factor for executive functions. Thus, a higher score indicates worse performance.

To evaluate executive functions, the interference score from the Color-Word Interference Test by J. R. Stroop was used [47]. The test consists of three subscales: name color, read color word, and interference. For each subscale, three rounds are given, which sums up to nine processing sheets. Time to completion is measured for each subscale in seconds, and the median was chosen for calculation. Thus, a higher score indicates worse performance. Consistency and retest-reliability are between *r* = 0.90 and *r* = 0.98. Validity studies revealed that interference scores are associated with selectivity, resistance to interference, and filter capability [47]. Furthermore, interference scores assess response inhibition [2] and executive functions [6,7,48].

The revised version of the d2 Test of Attention (d2-R) by Brickenkamp et al. [49] measures the individual attention and concentration performance and accuracy while differentiating similar visual stimuli.

The Multiple-Choice Vocabulary Test (MWT-B) is a multiple-choice vocabulary word test measuring the premorbid intelligence quotient [50]. The test consists of 37 rows of five words each, whereby only one word makes sense. The other four words are similar in both ways of writing as well as pronunciation of the real word. The word rows are organized by increasing difficulty. Participants can take as long as they need to process the test, but do not usually need more than five minutes. The version MWT-B, which was used in the current study, correlates highly with the MWT-A by *r* = 0.84. The retest-reliability after 14 months is *r* = 0.87. The MWT-B correlates with common intelligence scales including the Hamburg-Wechsler-Intelligence Scale (*r* = 0.81).

A prospective 4-day food record was given to patients to complete at home on the 4 consecutive days after cognitive testing. Participants had to report the exact number of foods and beverages they consumed, as well as the time and location. The protocol depicts ongoing consumption with the advantages of light workload and low costs [51]. The completed food records were then evaluated with the “nut.s–nutritional.software” version 1.32.03 [52]. According to previous literature, the following variables/nutrients were chosen a priori for analyses: fat, protein, carbohydrates, vitamin B6, vitamin B9, vitamin B12, omega 3 fatty acids, variety, and diversity. Variety constitutes the amount of different food items consumed and diversity represents the amount of different food categories consumed. For example, a diversity score of 18 and a variety score of 38 mean that 38 different foods were chosen out of 18 different food groups [52]. Basal metabolic rate was evaluated using the formula of Schofield [51], in which age and gender were considered. Underreporting was then computed with the Goldberg-formula [51].

### 2.4. Statistical Analyses

To test whether individuals with BD and healthy controls differ in nutrient intake, a multivariate analysis of covariance (MANCOVAs) as well as a regression analysis were performed. Single factor, multivariate analyses of covariance were executed with the clusters macronutrients, vitamins, and nutritional behavior as dependent variables, and group as the independent variable. The group **macronutrients** contained the variables protein, fat, and carbohydrates, and **vitamins** consisting of vitamin B6, vitamin B9, and vitamin B12. A second MANCOVA was performed with **diversity, variety**, and **energy intake** (kcal). Furthermore, a multivariate analysis of covariance was computed with **omega 3 fatty acids** as the dependent variable and group as the independent variable. WHtR, BDI, and smoking habit were included as covariates in all analyses.

Differences within the sample in demographic variables were tested with *t*-tests and chi-square (Χ^2^) statistics. Differences between patients and controls in cognitive test scores were computed with MANCOVAs and ANCOVAs (controlling for BDI, WHtR, and smoking).

Multiple hierarchical regression analyses were executed for verbal learning and memory, Stroop interference, TM performance, and d2-R performance in individuals with BD (*n* = 56). Independent variables were micronutrients, macronutrients, and diversity. The variable micronutrients were computed using standardized *z*-statistics. *Z*-values of the variables vitamins B6, B9, and B12, as well as omega 3 were summarized into one variable. The covariates smoking (yes/no), WHtR, and BDI were included in analyses as a first step. “Enter” was used as a method in SPSS as Bühner and Ziegler [53] suggest preferring this method over all other methods of stepwise reduction. Prerequisites for regression analyses were evaluated using correlations for linearity between variables, histograms for normal distribution of error variance, scatterplots for homoscedasticity, Durbin-Watson test for autocorrelations, and variance inflation factor (VIF) as well as tolerance for multicollinearity. These prerequisites are met in all regression analyses. Outliers were excluded when individual scores deviated more than three standard deviations. Bonferroni correction was abstained from because officious use of it creates a needless loss of power and may increase Type II error rates [54,55]. However, for a better understanding of the statistical results, effect sizes are reported for all effects for both computed multiple univariate analyses of covariance and for multivariate effects. All statistical analyses were computed using IBM SPSS version 22.0.

## 3. Results

### 3.1. Sample Characteristics

*T*-tests revealed a significant difference between groups in smoking (yes/no), BMI, WHtR, BDI, HAMD, and YMRS (Table 1). Ordinate variables school and professional education were tested using the Mann–Whitney U Test and both revealed significant differences between groups. However, premorbid IQ was used instead, since school education (*r* = 0.41, *p* < 0.001) and professional education (*r* = 0.62, *p* < 0.001) correlated with premorbid IQ. As a result, for univariate and multivariate analyses concerning cognition, IQ, smoking, and BDI were included as covariates. In univariate and multivariate analyses concerning nutrition, WHtR and smoking were included as covariates. Regression analyses included the dichotomous variable smoking (yes/no) and the BDI score as covariates.

A chi-square test revealed more females in the control group (*χ*^2^(1) = 5.24, *p* = 0.022), hence all univariate and multivariate analyses were also performed with sex included as an additional independent variable.

Patients with BD performed worse in the d2-R (*F*(1/94) = 6.75, *p* = 0.011, *η*^2^ = 0.067) than HC. No multivariate group effect was found in TMT performance (*F*(2/93) = 2.30, *p* = 0.106, *η*^2^ = 0.047) and CVLT performance *F*(3/92) = 1.79, *p* = 0.154, *η*^2^ = 0.055), and no differences in the Stroop interference task (*F*(1/93) = 1.55, *p* = 0.216, *η*^2^ = 0.016) were found. The univariate results are listed in Table 2. WHtR was a significant confounder for CVLT performance, (*F*(3/92) = 3.54, *p* = 0.018, *η*^2^ = 0.104), Stroop performance (*F*(1/93) = 9.22, *p* = 0.003, *η*^2^ = 0.090) as well as performance in the d2-R (*F*(1/94) = 9.23, *p* = 0.003, *η*^2^ = 0.089). The other control variables smoking and BDI had no effect on cognitive performance (*p* > 0.05).

Regarding macronutrients (protein, fat, and carbohydrate intake), a MANCOVA yielded no significant main effects for neither group (*F*(3/96) = 1.94, *p* = 0.129, *η*^2^ = 0.057) nor any of the covariates WHtR, smoking, and BDI (*p* > 0.05). Univariate results showed a trend in carbohydrate intake, indicating more carbohydrate consumption in the BD group (see Table 2). The MANCOVA testing differences in omega 3 fatty acids showed no significant group effect (*F*(3/96) = 1.27, *p* = 0.288, *η*^2^ = 0.04); neither of the co-variables were significant (*p* > 0.05, see Table 2).

A MANCOVA indicated that there was no difference in diversity, variety, and energy intake (kcal) between patients with BD and controls (*F*(3/96) = 1.40, *p* = 0.249, *η*^2^ = 0.04). Smoking had a significant effect on all three parameters (*F*(3/96) = 3.31, *p* = 0.023, *η*^2^ = 0.09).

With a confidence interval of 95%, 46.4% of bipolar patients and 56.6% of HC seemed to have underreported food intake. Using an even more conservative confidence interval of 99%, 76.8% of patients and 84.9% of controls reported too little consumption. However, chi-square test showed that groups did not differ in underreporting (95%: *χ*^2^ = 1.13, ns.; 99%: *χ*^2^ = 1.15, ns.)

### 3.2. Cognition and Nutrition

#### 3.2.1. Verbal Learning and Memory

Multiple hierarchical regression analyses revealed a significant effect for CVLT trial 1–5 (Model 1: *F*(3/96) = 5.54, *p* = 0.002; Model 2: *F*(8/91) = 3.30, *p* = 0.002), CVLT short delay free recall (Model 1: *F*(3/95) = 3.55, *p* = 0.017; Model 2: *F*(8/90) = 2.58, *p* = 0.014), and CVLT long delay free recall (Model 1: *F*(3/96) = 2.34, *p* = 0.078; Model 2: *F*(8/91) = 1.78, *p* = 0.092), however, no associations between CVLT parameters and nutritional behavior were shown. WHtR was the only significant parameter in the model showing associations with the CVLT scores. Table 3 shows the statistical values.

#### 3.2.2. Trail Making Performance

Multiple hierarchical regression analyses revealed no significant associations between TMT A/B and nutritional parameters. Both model 1 (*F*(3/96) = 3.69, *p* = 0.014) and model 2 (*F*(8/91) = 2.02, *p* = 0.053) showed a significant association between WHtR and TMT A performance (see Table 4). Analyses for TMT B did not yield any significant results (*F*(3/95) = 2.39, *p* = 0.073, *F*(8/90) = 0.960, *p* = 0.472).

#### 3.2.3. Stroop Interference

Multiple hierarchical regression analysis revealed significant results in both steps of the regression analysis (Model 1: *F*(3/94) = 4.83, *p* = 0.004; Model 2: *F*(8/89) = 2.38, *p* = 0.023). Results showed that the covariate WHtR had a high effect on Stroop interference performance, whereas smoking and BDI had no significant influence. In the second step of hierarchical regression analysis, a statistical trend of macronutrients on the Stroop interference task (*p* = 0.079) was observed. The regression coefficients are depicted in Table 5.

#### 3.2.4. D2 Attention Performance

Results revealed a significant first step of hierarchical regression analysis showing significant effects of WHtR and BDI on d2 performance (*F*(3/94) = 4.83, *p* = 0.004). The second model was also significant (*F*(8/89) = 2.38, *p* = 0.023) and showed a trend towards significance of diversity (*p* = 0.097). No significant results concerning micronutrients and macronutrients were found (see Table 6).

## 4. Discussion

The aim of this study was to analyze individuals with BD and HC regarding differences in nutrition (micro- and macronutrients) and cognition (attention, memory, executive function). Furthermore, the relationship between the average intake of micro- and macronutrients as well as nutritional diversity and different cognitive variables was investigated. Results showed no differences in nutritional behavior in individuals with BD and HC. Regarding cognition, individuals with BD showed significantly worse scores in the d2 test. Hierarchical regression analyses showed no association between cognitive and nutritional parameters, while WHtR was negatively correlated with cognitive performance in the domains.

The previously shown worse cognitive performance of individuals with BD in comparison to HC [1,2,5,6,8,10,56] could only partly be supported with the results of this study: individuals with BD performed worse in the d2-R, measuring attention and concentration [3], while tests assessing executive function and verbal memory showed non-significant trends in this direction. However, in contrast to other studies [13,14], we controlled for overweight by including WHtR as a covariate in statistical analyses. This factor has been found to affect cognitive parameters negatively [16,18], as shown by our results indicating WHtR as a significant confounder as well as a predictor of several cognitive test scores. This supports the notion that metabolic parameters, such as the WHtR, seem to be involved in the impairment of cognition that has often been observed in individuals with BD.

Concerning nutritional differences between individuals with BD and HC, none were found in this study, as opposed to previous literature indicating that individuals with BD seem to have an unhealthy diet [12,20,24]. Several reasons could explain these findings. Firstly, both groups might indeed not differ in their nutritional habits, because there might be other parameters negatively influencing the health and cognition of individuals with BD, such as medication intake and physical activity, both of which were not considered in this study. Indeed, a previous study with a partly overlapping cohort revealed that physical activity plays an important role in this relationship [57]. It is known that smoking affects the body’s ability to absorb and increase the turnover of a variety of vitamins and minerals, which might result in a lower concentration of micronutrients [58]. Smoking was significantly more frequent in individuals with bipolar disorder (around 50% were smokers) and might influence the processing of nutrients, resulting in lower availability of vitamins and minerals in individuals with BD despite similar eating habits. However, WHtR strongly influenced cognitive performance and was significantly higher in individuals with BD. It should thus not be discounted that nutrition could have contributed to this development.

Secondly, both groups might only subjectively have similar dietary habits, thus highlighting our issues concerning data collection. Studies involving self-observation might encourage heightened awareness of a specific behavior. Merely observing this behavior might change it, leading to a healthier diet, even if only during the time of the observation period. Moreover, participants may have not been completely honest in their observations, perhaps due to social desirability. This is supported by the fact that both groups underreported their food intake, although they did so to the same degree. These findings raise the question of whether another method for assessing daily food intake should be found. For example, the methods of 24-h dietary recall [59] conducted by a dietitian or weighted food record [60] might have led to a more accurate coverage of the sample’s food intake.

Thirdly, both groups might only majorly differ in their nutritional habits during an affective episode. This is supported by the fact that mostly euthymic individuals were included in this study. As cognition is impaired during an episode [61], the dynamic between cognition and nutrition may change as well, perhaps contributing to the explanation of non-significant correlations between the two areas of interest. Importantly, most cross-sectional studies found an association between cognitive impairment and number of episodes, whereas there is no clear evidence of the progress of cognitive decline in longitudinal studies [62]. Morevover, bipolar specific factors might play a role, as individuals with BD II might exhibit different cognitive deficits compared to individuals with BD I [56]. In our sample, two out of three of patients were diagnosed with BD I, which might have influenced the results.

Nutritional parameters did not predict the cognitive functioning of individuals with BD. Research on this topic is scarce and results seems to be inconsistent [63]. A review of Olagunju et al. [64] found that nutritional interventions were associated with improvement in cognitive domains, however, it was pointed out that the quality of said study was not ideal, mainly due to small sample sizes. In the present study, the sample size was limited as well, thus possibly failing to detect any significant associations, and there was no specific dietary intervention. Moreover, the duration of the observation period was short, although this period was intended to be representative of the individuals’ dietary habits. Nevertheless, it is recommended to conduct studies of greater dimensions, both in time as well as sample size.

## 5. Limitations

Several limiting factors were found. First, the cross-sectional design did not allow the determination of causality. Secondly, medication intake as well as physical activity were not taken into account. Thirdly, the sample size was rather small, and the observation period was relatively short, possibly leading to the inability to detect significant differences in nutritional behavior. Finally, the results might have been influenced by the method of assessing daily food intake and should be interpreted accordingly.

## 6. Conclusions

Individuals with BD show impaired cognitive function (attention and concentration) compared to HC. High WHtR seems to be an especially important factor, negatively influencing cognition in BD. Since there is a profound lack of research on nutritional differences and their impact on cognition, this relationship should be examined in further studies involving objective measurements to assess food intake as well as adequate sample sizes.

## Figures and Tables

**Table 1 nutrients-14-01176-t001:** Sample characteristics.

Variable	Bipolar Patients (*n* = 56)	Healthy Controls (*n* = 53)		
	*M (SD)*	*M (SD)*	Test Statistics (*t*, *χ*^2^)	*p*-Value
Age, mean (*M*, *SD*)	39.78 (11.29)	37.03 (12.85)	1.19	0.236
Sex, *n* (%)				
Female	27 (48.2%)	37 (69.8%)	5.24	**0.022** *
Male	29 (51.8%)	16 (30.2%)		
Bipolar I/Bipolar II	37/18			
Premorbid IQ, mean (*M*, *SD*)	112.15 (14.88)	113.76 (14.74)	−0.55	0.581
Smoking yes/no	27/29	11/42	9.04	**0.003** **
BMI (*M*, *SD*)	27.68 (6.36)	24.42 (4.45)	3.12	**0.002** **
Normal weight (%)	37.5%	66.0%		
Overweight (%)	32.1%	18.9%		
Obese (%)	30.4%	15.1%		
WHtR, mean (*M*, *SD*)	0.54 (0.09)	0.49 (0.07)	3.15	**0.002** **
Normal (%)	35.2%	60.8%		
Risk (%)	64.8%	39.2%		
BDI, mean (*M*, *SD*)	13.68 (11.08)	3.30 (3.24)	6.55	**<0.001** **
HAMD, mean (*M*, *SD*)	5.27 (4.24)	0.22 (0.94)	8.50	**<0.001** **
YMRS, mean (*M*, *SD*)	1.16 (3.60)	0.00 (0.00)	2.41	**0.019** *

Note: BMI = body mass index; WHtR = waist to height ratio; BDI = Beck Depression Inventory; HAMD = Hamilton Depression Scale; YMRS = Young Mania Rating Scale; SD = standard deviation, * and in bold: statistically significant at *p* < 0.05, ** and in bold: statistically significant at *p* < 0.01.

**Table 2 nutrients-14-01176-t002:** Cognitive scores and nutrients intake in bipolar patients versus healthy controls.

	Bipolar Patients	Healthy Controls			
	*M (SD)*	*M (SD)*	Test Statistic (*F*)	*p*-Value	*η* ^2^
**Cognitive test scores**					
TMT A (s)	32.00 (10.37)	26.52 (8.73)	4.65	**0.034** *	0.047
TMT B (s)	72.68 (28.34)	60.00 (23.86)	1.36	0.247	0.014
TMT B–TMT A (s)	39.68 (25.01)	33.45 (19.08)	0.201	0.655	0.002
Stroop interference (s)	75.23 (15.55)	68.09 (12.22)	1.55	0.216	0.016
CVLT *sum trial 1–5*	56.90 (10.76)	61.82 (9.84)	0.524	0.471	0.006
CVLT *short delay free recall*	11.80 (2.91)	13.19 (2.42)	3.50	0.064	0.036
CVLT *long delay free recall*	12.71 (2.94)	13.63 (2.48)	0.928	0.338	0.010
D2 Test of Attention	158.33 (39.60)	200.38 (51.60)	6.75	**0.011** *	0.067
**Nutrients intake**					
Protein (g)	77.91 (24.71)	70.86 (21.49)	2.40	0.125	0.024
Fat (g)	80.66 (32.68)	80.36 (21.91)	0.42	0.518	0.004
Carbohydrates (g)	214.05 (85.03)	189.62 (57.79)	3.78	0.055	0.037
Vitamines B12 (µg)	4.89 (3.29)	4.37 (2.43)	0.008	0.929	0.000
Vitamines B6 (µg)	1484.98 (491.43)	14,49.24 (626.71)	0.464	0.497	0.005
Vitamines B9 (µg)	236.78 (83.71)	246.32 (87.70)	2.26	0.136	0.022
Vitamine D (µg)	1.96 (1.40)	2.29 (1.58)	0.002	0.968	0.000
Omega 3 fatty acids	1658.33 (1052.86)	1723.32 (761.33)	0.160	0.690	0.002
Omega 6 fatty acids	12,764.02 (5335.40)	12,923.29 (4673.61)	0.277	0.600	0.003
Poly-saturated fatty acids	14,433.35 (5946.37)	14,541.96 (5515.91)	0.300	0.585	0.003
Diversity	10.35 (2.47)	11.39 (2.55)	0.444	0.507	0.005
Variety	27.96 (7.94)	32.53 (11.59)	1.60	0.209	0.016
Energy intake (kcal)	1935.32 (635.59)	1892.54 (8554.52)	1.78	0.185	0.018

Note: results of univariate and multivariate analyses of co-variance (controlled for waist to height ratio, smoking and Beck Depression Inventory) testing differences between bipolar patients and healthy controls. TMT = Trail Making Test, CVLT = California Verbal Learning Test, * statistically significant at *p* < 0.05.

**Table 3 nutrients-14-01176-t003:** Association of nutritional behavior with CVLT parameters.

		CVLT Trial 1–5			CVLT Short Delay Free Recall			CVLT Long Delay Free Recall		
		β	*t*	*p*	β	*t*	*p*	β	*t*	*p*
Model 1	Smoking	−0.01	−0.014	0.890	0.03	0.33	0.743	−0.18	−0.18	0.861
	WHtR	−0.33	−3.44	**0.001** **	−0.31	−3.12	**0.002** **	−0.25	−2.52	**0.013** *
	BDI	−0.14	−1.40	0.166	−0.05	−0.47	0.637	−0.02	−0.19	0.847
Model 2	Smoking	0.02	0.15	0.885	0.10	0.92	0.363	0.04	0.32	0.753
	WHtR	−0.31	−3.11	**0.003** **	−0.30	−2.88	**0.005** **	−0.23	−2.24	**0.028** *
	BDI	−0.16	−1.53	0.129	−0.05	−0.44	0.660	−0.01	−0.12	0.902
	Micronutrients	0.23	1.57	0.120	0.22	1.39	0.168	0.19	1.21	0.230
	Protein	0.07	0.44	0.664	0.04	0.23	0.820	0.02	0.09	0.929
	Fat	−0.24	−1.56	0.121	−0.04	−0.28	0.778	−0.06	−0.38	0.705
	Carbohydrates	−0.16	−1.37	0.174	−0.19	−1.91	0.112	−0.18	−1.49	0.139
	Diversity	0.12	1.13	0.263	0.18	1.57	0.120	0.17	1.48	0.144

Note: **CVLT trial 1–5:** Model 1: R = 0.38, R^2^ = 0.15, R^2^corr = 0.12, SE = 9.58; Model 2: R = 0.47, R^2^ = 0.23, R^2^corr = 0.16, SE = 9.65; **CVLT short delay free recall:** Model 1: R = 0.32, R^2^ = 0.10, R^2^corr = 0.07, SE = 2.66; Model 2: R = 0.43, R^2^ = 0.17, R^2^corr = 0.11, SE = 2.60; **CVLT long delay free recall:** Model 1: R = 0.26, R^2^ = 0.07, R^2^corr = 0.04, SE = 2.69; Model 2: R = 0.37, R^2^ = 0.16, R^2^corr = 0.06, SE = 2.66. Significant *p*-values are written in bold and marked with * (<0.05) or ** (<0.01).

**Table 4 nutrients-14-01176-t004:** Association of nutritional behavior with TMT parameters.

		TMT A			TMT B		
		β	*t*	*p*	β	*t*	*p*
Model 1	Smoking	0.028	0.272	0.786	0.077	0.729	0.112
	WHtR	0.259	2.63	**0.010** *	0.162	1.60	0.197
	BDI	0.138	1.33	0.187	0.138	1.39	0.146
Model 2	Smoking	0.039	0.359	0.721	0.074	0.650	0.517
	WHtR	0.304	2.93	**0.004** **	0.180	1.66	0.100
	BDI	0.147	1.39	0.169	0.142	1.28	0.202
	Micronutrients	−0.069	−0.447	0.656	−0.017	−0.105	0.916
	Protein	−0.197	−1.11	0.270	−0.024	−0.131	0.896
	Fat	−0.012	−0.077	0.939	−0.054	−0.328	0.743
	Carbohydrates	0.142	1.19	0.239	−0.025	−0.200	0.842
	Diversity	0.118	1.03	0.308	0.055	0.459	0.647

Note: **TMT A:** Model 1: R = 0.32, R^2^ = 0.10, R^2^corr = 0.08, SE = 9.67; Model 2: R = 0.39, R^2^ = 0.15, R^2^corr = 0.08, SE = 9.67; **TMT B:** Model 1: R = 0.27, R^2^ = 0.07, R^2^corr = 0.04, SE = 26.24; Model 2: R = 0.28, R^2^ = 0.08, R^2^corr = −0.00, SE = 26.94. Significant *p*-values are written in bold and marked with * (<0.05) or ** (<0.01).

**Table 5 nutrients-14-01176-t005:** Association of nutritional behavior with Stroop interference.

		Stroop Interference		
		β	*t*	*p*
Model 1	Smoking	−0.015	−0.150	0.881
	WHtR	0.331	3.38	**0.001**
	BDI	0.119	1.16	0.249
Model 2	Smoking	−0.029	−0.271	0.787
	WHtR	0.279	2.72	**0.008**
	BDI	0.126	1.21	0.232
	Micronutrients	−0.274	−1.78	0.079
	Protein	0.238	1.36	0.179
	Fat	0.001	0.007	0.995
	Carbohydrates	0.110	0.918	0.361
	Diversity	−0.047	−0.408	0.685

Note: **Stroop interference:** Model 1: R = 0.37, R^2^ = 0.13, R^2^corr = 0.12, SE = 13.61; Model 2: R = 0.42, R^2^ = 0.18, R^2^corr = 0.10, SE = 13.64. Significant *p*-values (*p* < 0.05) are written in bold.

**Table 6 nutrients-14-01176-t006:** Association of nutritional behavior with d2 attention performance.

		D2 Attention Performance		
		β	*t*	*p*
Model 1	Smoking	−0.030	−0.311	0.756
	WHtR	−0.339	−3.65	**<0.001**
	BDI	−0.236	−2.41	**0.018**
Model 2	Smoking	0.047	0.470	0.640
	WHtR	−0.300	−3.10	**0.003**
	BDI	−0.234	−2.38	**0.019**
	Micronutrients	0.241	1.66	0.100
	Proetein	−0.246	−1.48	0.143
	Fat	0.056	0.380	0.705
	Carbohydrates	0.033	0.293	0.770
	Diversity	0.178	1.68	0.097

Note: **d2 attention performance:** Model 1: R = 0.37, R^2^ = 0.13, R^2^corr = 0.11, SE = 13.61; Model 2: R = 0.42, R^2^ = 0.18, R^2^corr = 0.10, SE = 13.64. Significant *p*-values (*p* < 0.05) are written in bold.

## Data Availability

The data presented in this study are available on request from the corresponding author. The data are not publicly available due to ethical reasons.

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
