# Peer review of "Nutrition, Overweight, and Cognition in Euthymic Bipolar Individuals Compared to Healthy Controls"

_nutrients, 2022, doi:10.3390/nu14061176_

Round 1

Reviewer 1 Report

The manuscript "Nutrition, Overweight, and Cognition in Euthymic Bipolar Individuals compared to Healthy Controls" deals with a current and interesting topic. It presents data on impairment in cognitive function, overweight and nutritional habits. Furthermore, the study investigate the relationship of intake of micro- and macronutrients, nutritional diversity and cognitive variables.  Unfortunately, the data on physical activity and medication is missing and the sample size is rather small. But the limitations are reported and discussed adequatly. The manuscript is well written.  

Author Response

Thank you very much for your feedback on our paper. If we understand it right are no further changes necessary? Thank you very much for your efforts!

Reviewer 2 Report

The present work with cognitive alterations in bipolar disorder and their association with diet. Overall, the topic is interesting for clinicians, the study is well conducted, and the manuscript is well written. However, I have some questions for the authors:

- The cohort includes 56 euthymic individuals with BD. But it is mentioned that some patients are hospitalized. Therefore, does the study include patients with different severity condition or what is the reason to hospitalize euthymic patients? Can the hospitalization affect the diet? (In hospitals the diet could be different/healthier).

- There is an error in table 1: in the bipolar column, smoking (yes/no) 297/29?

- Differences between patients and HC are important, and I think that this fact could bias the results? The authors only talk about medication but there are other variables such as smoking that could be important.

- Cross-sectional studies find an association between cognitive impairment and the number of episodes, while longitudinal studies do not seem to explain the deficits. The authors did not consider this aspect and it need to be discussed (Solé et al., 2017. International Journal of Neuropsychopharmacology (2017) 20(8): 670-680).

- Moreover, cognitive impairments seem to be less severe in BDII than in BDI. Can the authors better describe the sample and discuss this issue (Bora, Yücel, Pantelis, & Berk, 2011. Acta Psychiatrica Scandinavica, 123, 165-174).
